# Characteristics and Pathogenicity of *Discula theae-sinensis* Isolated from Tea Plant (*Camellia sinensis*) and Interaction with *Colletotrichum* spp.

**DOI:** 10.3390/plants12193427

**Published:** 2023-09-28

**Authors:** Qingsheng Li, Junyan Zhu, Ning Ren, Da Li, Ya Jin, Wenyuan Lu, Qinhua Lu

**Affiliations:** 1Institute of Sericulture and Tea, Zhejiang Academy of Agricultural Sciences, Hangzhou 310021, China; liqs@zaas.ac.cn (Q.L.); renn@zaas.ac.cn (N.R.); lida@zaas.ac.cn (D.L.); jinya202204@163.com (Y.J.); 2State Key Laboratory of Tea Plant Biology and Utilization, Anhui Agricultural University, Hefei 230036, China; zhujy@ahau.edu.cn; 3College of Horticulture Science, Zhejiang A&F University, Hangzhou 311300, China; 4Development Center of Agricultural Science and Technology in Huzhou, Huzhou 313000, China; tea66@126.com

**Keywords:** *Camellia sinensis*, co-inoculation, *Colletotrichum*, *Discula theae-sinensis*, pathogenicity

## Abstract

Anthracnose is one of the primary diseases in tea plants that affect tea yield and quality. The geographical distribution, occurrence regularity, and agronomic measures of tea plants with anthracnose have been researched for decades. However, the pathogenic cause of anthracnose in tea plants is diverse in different regions of the world. Identifying the specific pathogenic fungi causing tea anthracnose is an essential control measure to mitigate this disease. In this study, 66 *Discula theae-sinensis* and 45 *Colletotrichum* isolates were obtained from three different types of diseased tea leaves. Based on multilocus phylogenetic and morphological analysis, eight known species of *Colletotrichum*, *Colletotrichum fructicola*, *C. camelliae*, *C. aenigma*, *C. siamense*, *C. henanense*, *C. karstii*, *C. tropicicola*, and *C. gigasporum* were identified. This study is the first to report *C. tropicicola* and *C. gigasporum* in tea plants in China. *Discula theae-sinensis* was the most common species in this study and caused disease lesions around wounded areas of tea leaves. The dual trials in vitro indicated *Discula theae-sinensis* and *Colletotrichum* were slightly inhibited. Co-inoculating *Discula theae-sinensis* and *C. fructicola* was superior to single inoculation at low concentrations. The main cause of anthracnose might be the concerted action of a variety of fungi.

## 1. Introduction

Tea, which is produced from the new shoots from tea plants [*Camellia sinensis* (L.) O. Kuntze] is a globally consumed beverage. The production of tea includes cultivation, processing, packaging, and exportation, and is distributed across Asia, Africa, South America, and Oceania. The tea industry is a high-output and distinctive agriculture industry in China [1]. However, fungal diseases pose a threat to both the yield and quality of tea. Anthracnose, a severe and widespread disease, significantly affects the growth of tea plants and reduces the quality of tea products [2]. Anthracnose primarily occurs in warm and moist regions, with the peak diffusion period typically spanning from April to July, contingent on climatic conditions and regional distribution [3].

To date, the identification of pathogens responsible for anthracnose in tea plants in China remains a subject of controversy. Over the past several decades, the pathogen causing anthracnose was recognized as *Gloeosporium theae-sinensis* [4]. Nevertheless, the taxonomy of *G. theae-sinensis* is ambiguous. Yamamoto proposed that *G. theae-sinensis* represented an invalid recombination of *C. theae-sinensis*, which should be classified with the *Colletotrichum* genus [5]. However, Moriwaki et al. considered that *G. theae-sinensis* should be designated *Discula theae-sinensis* (Dt), which aligned with *Diaporthales* in terms of morphological and molecular characterization [6]. *Diaporthales*, an order within the subclass *Sordariomycetes*, has the potential to induce anthracnose in various crops and trees [7,8,9]. *Diaporthales* also widely exist in tea plants as pathogens or endophytes. Xie et al. summarized numerous studies on endophytes in tea plants and listed three families of *Diaporthales* reported as endophytes in tea plants, namely *Diaporthaceae*, *Melanconiellaceae*, and *Valsaceae* [10]. Gao et al. investigated *Diaporthe* (*Diaporthecae*, *Diaporthale*) species associated with symptomatic and asymptomatic tissues of *Camellia*, identifying four novel species, three known species, and two complex species from diseased and healthy tissues of tea plants from several provinces in China [11]. Nevertheless, their distribution in China and pathogenicity remain unclear.

In addition to Dt, *Colletotrichum* is considered another major cause of anthracnose in tea plants in China. *Colletotrichum* is a large genus of Ascomycete fungi belonging to *Glomerellales* [12]. Via sampling in multiple regions, 21 species and one indistinguishable strain of *Colletotrichum* were reported to infect tea plants in China [13,14]. Based on geographic distribution and strain quantity, *Colletotrichum camelliae* (Cc) and *C. fructicola* (Cf) were identified as the principal causal agents of tea anthracnose [13,15]. Cc probably hosts specific taxa that occur on *Camellia*, while Cf has a more diverse host range [13]. In terms of pathogenicity, Cc exhibits greater virulence than Cf on tea plants. Furthermore, secondary metabolites in tea plants such as catechins and caffeine influence the virulence of pathogens [16]. However, Cc and some *Colletotrichum* species are also supposed to be pathogens of tea leaf blight [17,18]. Consequently, it is essential to definitively ascertain the primary cause of tea plant anthracnose in China.

As the predominant causal agents of plant diseases, fungi employ diverse strategies to colonize plants and induce diseases [19]. Generally, different fungi utilize distinct mechanisms to penetrate host plants. Dt enters young tea leaves solely through trichomes [20,21]. Furthermore, Dt is the sole fungus that enters the host plant only through the trichomes [22]. *Colletotrichum* spp. typically develop appressoria to breach the host’s cuticle layer and cell wall [23,24]. Appressoria originates from conidial germ tubes and secrete effectors via appressorial penetration pores [25,26]. Host plants must employ specific defense strategies against various fungi. Thus, identifying the primary microorganisms responsible for tea diseases can enhance the effectiveness of disease control in tea plants.

In the present study, we isolated pathogens and collected anthracnose-diseased tea leaves over two years to elucidate the primary cause of anthracnose in tea plants in China. We selected the different types of diseased tea leaves based on field phenotypes and employed a combination of multilocus phylogeny and morphological characteristics to identify the species of all isolates. Out of the isolates, 66 were identified as Dt, and 45 were *Colletotrichum*. Dt and Cf were commonly found in *Diaporthales* and *Colletotrichum*, respectively. Additionally, pathogenicity tests were conducted on representative Dt isolates and co-inoculated with Dt and *Colletotrichum* to explore the primary cause of tea plant anthracnose in China.

## 2. Results

### 2.1. Phenotypic Characterization of Collected Tea Leaves

We collected diseased tea leaves with visible anthracnose symptoms continuously for two years from June to August. Based on detailed phenotypic differences in details, the symptoms of disease samples can be classified into three types. The first pattern (P1) displayed white disease spots accompanied by tiny black particles on the surface, which only existed on one side of the blade. The scabs of the second pattern (P2) were tawny with scorching on the side of the leaves. In the third pattern (P3), lesions were distributed on the distichous sides at the tip of the blade. The commonality among these three groups was water-soaked lesions close to the demarcation between diseased and healthy parts, which appeared translucent when exposed to light, and no ring spots (Figure 1A).

Based on microscopic examination, the forms of the collected acervuli from diseased tea leaves could be divided into two kinds (Figure 1B). The first kind of acervuli (A1) was suborbicular and flat, which constituted the main type among diseased leaves. The second kind of acervuli (A2) was irregular in shape with a split at the center. The A2 type only appeared once during procedures of sample collection procedures in this study. Additionally, the dominant spores on the scab were divided into two types (Figure 1C). One type was fusiform to obovoid, tapered at the base or both ends (S1), and the other type was cylindrical with obtuse ends (S2). Based on their morphological characteristics, we preliminarily considered S1 to be *Diaporthales* spp. and S2 to be *Colletotrichum* spp. Further studies revealed that all isolates collected from type A2 diseased tea leaves were identified as Cf.

### 2.2. Identification and Characterization of Dt

In our study, we collected 66 Dt isolates. Phylogenetic trees (Figure 2) were constructed using the combined ITS, EF1, RPB2, and LSU gene data, encompassing 76 *Diaporthales* strains, including *Valsella salicis* AR3514 as the outgroup. The combined gene alignment, including gaps, comprised 2248 characters with 579, 518, 247, and 893 bases having alignment gaps in ITS, EF1, RPB2, and LSU. Sixty-six isolates clustered with strain MAFF238240-238143. Dt was the most common species, with 62 isolates separated from the diseased part and four isolates from the healthy part.

Based on phylogenetic analysis, the cluster of DX12, DX14, DX15, and DX16 had a higher expected number of changes per site. The pairwise dissimilarities of DNA sequences between DX12, DX14, DX15, DX16, and MAFF238240 were 23 bp, 23 bp, 22 bp, and 21 bp in LSU, respectively.

***Discula theae-sinensis*** (I. Miyake) Moriwaki and Toy. Sato, J. Gen. Plant Pathol., 75:359–361. 2009.

*Discula thea-sinensis* are low, dense, fluffy white aerial mycelia. In reverse, they are concentric circles (DX7) and some are a uniform gray-white (DX14) (Figure 3). The growth rate of DX7, DX14, and DX44 was 2.63 and 2.40 per day at 25 °C on the PDA plate. The α-conidia of these three strains were hyaline, fusiform to obovoid, and tapered at the ends. The size of DX7 was 4.2–6.2 × 1.9–3.2 µm, average (av) ± SEM = 5.0 ± 0.6 × 2.4 ± 0.4 µm, n = 30. The size of DX14 was 3.2–6.3 × 1.7–2.8 µm, av ± SEM = 4.9 ± 0.7 × 2.2 ± 0.3 µm, n = 30. Moriwaki named this fungus Dt based on morphology and molecular characteristics, and the reported size of conidia of Dt is 4.6–6.3 × 1.9–3.1 µm. Conidia of strain DX7 were similar to the strain Moriwaki et al. reported, and DX14 was smaller than that strain.

### 2.3. Identification and Characterization of Colletotrichum spp.

Among all collected isolates, 45 isolates belonged to *Colletotrichum*. Phylogenetic trees (Figure 4) were constructed using the combined ITS, CAL, TUB2, and GAPDH gene data. The concatenated dataset contained 210 sequences from the genus *Colletotrichum* including outgroup *C. xanthorrhoeae* (CBS 127831), and 541, 744, 721, and 270 bases had alignment gaps in ITS, CAL, TUB2, and GAPDH, respectively. The isolates in *Colletotrichum* collected in the present study belonged to eight known species, Cf, Cc, *C. aenigma*, *C. siamense*, *C. henanense*, *C. karstii*, *C. tropicicola*, and *C. gigasporum*. The isolates in our study clustered with high posterior probability values ≥0.95. Twenty-seven isolates clustered with Cf, 11 isolates clustered with Cc, 2 isolates clustered with *C. aenigma*, and 1 isolate clustered with *C. siamense*, *C. henanense*, *C. karstii*, *C. tropicicola*, and *C. gigasporum.* Cf and Cc were the largest numbers of pathogens, accounting for 60% and 24.4% of all *Colletotrichum* strains, respectively, and the other six species constituted 15.6% of the total.

***Colletotrichum tropicicola*** S. Phoulivong, P. Noireung, L. Cai and K.D. Hyde, Crytog. Mycol. 33: 356. 2012.

*C. tropicicola* was first reported on leaves of *Citrus maxima. C. tropicicola* was first reported in tea plants, which are putative endophytes in the present study. The strain CX33 had a growth rate of 8.57 mm per day at 25 °C. The colony on the PDA was flat with complete edges, aerial mycelia that were short and dense, with orange conidial masses near the center, and white with gray pigment spots on the reverse side near the center after 7 days (Figure 5A). Conidia on the PDA were hyaline, guttulate, long elliptic with obtuse ends, measuring 11.1–18.5 × 4.8–7.1 µm, av ± SEM = 15.0 ± 0.3 × 5.8 ± 0.1 µm, n = 30. conidia dimensions of CX33 were smaller than ex-type culture (LC0598 15–19 × 6–7 μm). Appressoria on SNA (7.2–16.4 × 6.2–11.8, av ± SEM = 12.0 ± 2.5 × 8.2 ± 1.5 µm, n = 30) were black, elliptical, ovoid, or slightly irregular.

***Colletotrichum gigasporum*** E.F. Rakotoniriana and F. Munaut, Mycol. Progr. 12: 407. 2013.

In the present study, *C. gigasporum* was first reported in tea plants. The colony on the PDA was flat with smooth edges, with white pigmentation becoming black toward the center, and in reverse, typically exhibited white-to-black annular pigment after 7 days (Figure 5B). The growth rate was 11.49 mm per day. The conidia on the PDA (19.9–27.2 × 5.6–8.4 µm, av ± SEM = 23.3 ± 0.3 × 7.1 ± 0.1 µm, n = 30) were hyaline, aseptate, smooth, cylindrical with round ends. *C. gigasporum* is characterized by large conidia (22–)25–29(–32) × (6–)7–9 μm [27]. CX34 conforms to large conidia characteristics. Appressoria on SNA (11.8–16.3 × 9.3–13.6, av ± SEM = 14.4 ± 1.2 × 11.1 ± 1.0 µm, n = 30) were brown to black, ovoid, or irregular.

### 2.4. Pathogenicity Test of Dt Collected from Leaf Surface

To explore the virulence of Dt strains collected in the present study, we used mycelial discs of representative isolates of Dt (DX7, DX14) to infect the third tea leaves and carry out pathogenicity assays. The results are shown in Figure 6. Symptoms appeared after inoculation. All these isolates caused necrotic lesions around the wounded areas. As for the details of lesions, suborbicular and flat acervuli can be found in the disease spot at 7 days. The traits of acervuli in inoculated tea leaves were similar to those in diseased leaves in the field.

### 2.5. Dual Inoculation Assays of Dt and Colletotrichum

As mentioned above, Dt and *Colletotrichum* co-existed in tea leaves. Dual trials in vitro were carried out to find a mutual effect between Dt and *Colletotrichum* (Figure 7). Cf and Cc were used to co-inoculate with Dt, respectively. For 5 days of dual growth, it was observed that inhibition percentages of Dt were less than those of *Colletotrichum.* As shown in Figure 7B, Dt and *Colletotrichum* maintained a short distance (<2 mm) between them. This performance indicated that Dt and *Colletotrichum* had a slight inhibition in vitro.

To investigate the relationship between them during infection, dual inoculation assays were applied (Figure 8). As shown in Figure 8A, the results of single inoculation were consistent with previous studies showing that Cc was more invasive than Cf. The PDA disc of Dt also caused tea leaf disease. Meanwhile, conidial suspensions Cf and Dt did not lead to obvious lesions on tea leaves. The results of dual inoculation were more intriguing (Figure 8B). The single inoculation and dual inoculation related to virulent Cc had similar results. Dts + Cc and Dtd + Cc gave rise to significantly dark lesions on the tea leaf surface. Nonetheless, a combination of Dts + Cf and Dtd + Cf enhanced pathogenicity. Compared to single inoculation, the dual inoculation of Cf and Dt resulted in larger lesions. The results indicated that the dual inoculation of Dt and Cf in vivo promotes the development of tea plant anthracnose.

### 2.6. Interaction of Dt and Cf on Tea Plant Infection

To explore the impact of fungal biomass on the synergistic effect, we inoculated tea leaves with different concentrations of spore suspensions of Dt and Cf, which were set to 10^5^, 10^6^, and 10^7^ spores/mL (Figure 9). The single inoculation results indicated that the spore suspension of Dt could not bring about obvious disease spots regardless of how many concentrations of spore suspensions we used. Lower concentrations of Cf (Cf5 and Cf6) caused limited damage to host plants, and the high concentration (Cf7) had the opposite effect, which generated apparent necrosis. As the lesion diameters showed, the lesion size of the F5 group (Dt5Cf5, Dt6Cf5, and Dt7Cf5) was significantly larger than those of Cf5 and Dt5, Dt6, and Dt7; the same applied for the F6 group (Dt5Cf6, Dt6Cf6, and Dt7Cf6) compared with those of Cf6 and Dt5, Dt6, and Dt7. We also found a significant difference in the F6 group. Compared to the single inoculation of Cf7, the F7 group (Dt5Cf7, Dt6Cf7, and Dt7Cf7) caused smaller lesions. Mixing with spore suspensions of low pathogenic Dt may reduce the virulence of highly pathogenic strains. The results showed that dual inoculation of Dt and Cf could promote pathogenicity when they were inoculated independently at lower concentrations. Moreover, the hypovirulent strain may recede the pathogenicity of the high-virulence strain when the pathogenicity of fungi used in dual inoculation differs greatly.

As Figure 1 shows, water-soaked lesions often occur on the anthracnose of tea plants in the field. During the procession of this study, we found that water-soaked lesions were around the fringe of necrosis when the inoculator was hypovirulent and formed a dispersed distribution of necrosis in the F5 and F6 groups. Especially in the F5 group, water-soaked lesions were in common. Furthermore, as a virulent strain, Cc also did not lead to noticeable water-soaked lesions (Appendix A). It was speculated that water-soaked lesions might play a role in affecting disease development. Exploring the composition and function of water-soaked lesions may be meaningful research in the future.

## 3. Discussion

### 3.1. Phenotype and Isolated Main Fungi of Diseased Tea Leaves

As a primary tea disease, anthracnose affects the yield and quality of tea [2,3]. According to previous studies, *Colletotrichum* is considered the dominant species that causes anthracnose in tea plants in China, while Dt is the major fungi that leads to tea anthracnose in Japan [5,18,20]. In the first year, we collected tea foliage with anthracnose and isolated both Dt and *Colletotrichum*. To verify the results, we repeated the experiment with new diseased tea leaves with anthracnose the next year, and both aforementioned fungi were detected as well. Thus, Dt widely exists in diseased tea plants as a pathogen. This is a new perspective that can guide research on anti-anthracnose strategies for tea plants with more accurate pathogen recognition.

Phenotypic identification of diseased leaves in the field is the primary method for determining specific diseases in tea plants. However, two typical diseases, anthracnose and tea leaf blight, are often easily confused, due to the complexity of environmental factors and the similarity of disease symptoms. Relying solely on visual observation for tea disease identification may lead to inaccuracies [28]. In this study, we collected pathogens from three different typical anthracnose symptoms, all characterized by sub-round or irregular scabs, scattered small black sporophores, water-soaked lesions, and the absence of thin ring spots. All three main fungi, Cc, Cf, and Dt, were found in topical anthracnose leaves collected in this study. Interestingly, all pathogenic isolates collected from A2 were Cf. This result serves as a notable example for investigating the relationship between specific pathogens and micro-characteristics such as acervuli form in plant disease.

In the present study, water-soaked lesions appeared on field samples and dual inoculation of Dt and Cf. During pathogen infection, water-soaking spots are frequently observed on infected leaves as an early symptom of the disease [29,30]. As time progresses, these spots enlarge and become necrotic [31]. In rice crops, water-soaked lesions with a dark green border are one of the field symptoms displayed by the fungi causing blast disease [32]. Water-soaking and necrosis in leaves infected by *Xanthomonas euvesicatoria* and *X. gardneri* can promote colonization of *Salmonella enterica* inside plant tissues [33]. Water-soaking symptoms may be related to water availability in the apoplastic environment, affecting microbial community structure and colonization [34]. AvrHah1, a single transcription activator-like (TAL) effector (TALE) protein of *X. gardneri*, confers enhanced water-soaked lesions in pepper [35]. Comparing the process of tomato infection with *X. gardneri* vs. *Xg*Δ*avrHah1*, water uptake from outside the leaf into the apoplast can be observed in *X. gardneri*-infected tomato, but not *Xg*Δ*avrHah1*. Water can facilitate bacteria entry into the apoplast [36]. As mentioned above, water-soaked spots affected plant–microbe interactions and might promote microbe entry. In our study, we found that water-soaked lesions always occurred in the weakly pathogenic infection process rather than in the highly pathogenic form. Therefore, we hypothesized that weak pathogens may utilize water-soaked spots to create a suitable living environment, in order to successfully infect host plants. Furthermore, we should conduct an in-depth study on the formation mechanism of water-soaked spots in tea plant–microbe interactions.

### 3.2. Identification of Dt from Tea Plant

Compared to the identification of *Colletotrichum* spp., determining the correct position of Dt in *Diaporthales* was more challenging. For a long time, species recognition principles in *Diaporthales* depended on culture characteristics, ascospore shape, and perithecium position. Currently, accurate DNA data coupled with conidial characteristics are used in scientific studies to identify and classify *Diaporthales.*

The nuclear ribosomal internal transcribed spacer (ITS) is commonly used when identifying the Kingdom of fungi. However, ITS sequences used alone in phylogenetic analysis are not ideal for closely related species due to intraspecific variation [37,38]. In previous studies, many researchers revised the order *Diaporthales* based on analysis of LSU nrDNA sequence data [7,39]. Moriwaki et al. selected 28S rDNA to study the phylogenetic relationships of strains MAFF 238,240 to 238,244 and *Colletotrichum* species [6]. Van Rensberg et al. [40] used ITS and EF1 sequence data to characterize species of *Phomopsis*. Hyde et al. [41] suggested that ITS and EF1 are recommended for preliminary identification of *Diaporthe* species. Senanayake et al. used combined DNA sequence data of ITS, EF1, and RPB2 to investigate the phylogenetic relationships of *Diaporthalean* families [42]. Thus, ITS, LSU, EF1, and RPB2 genes were used for sequence analysis in our study.

Dt strains collected in this study belong to *Diaporthales* species, these strains collected in this study fell in the same clade of MAFF 238,240 to 238,244 fungi with 100% pro. Moriwaki et al. used the LSU sequence alone to construct a phylogeny tree [6]. In our results, ITS, EF1, and RPB2 sequence data of *Discula theae-sinensis* are conducive to the following research. It is worth mentioning that the strains DX12, DX14, DX15, and DX16 had a higher expected number of changes per site. Combined with the characteristics of colony and α-conidial cells, we considered strains DX12, DX14, DX15, and DX16 to belong to Dt. The details of β- conidia or teleomorphs can provide more accurate identification for those species.

### 3.3. Interaction of Dt and Colletotrichum

In contrast to the type of host–pathogen interaction in monospecies infections, plants are exposed to various biotic challenges simultaneously in the field. Multiple pathogens may encounter the same host simultaneously [43]. Three pathogens were found in association with the black spot disease complex of peas [44]. Oomycete *Phytophthora sojae* and *Fusarium* spp. frequently coexist in diseased soybean roots, both of which cause soybean root rot [45]. In a two-year field investigation and sample collection, we discovered Dt, Cf, and Cc coexisting in diseased tea leaves.

Interactions between different pathogens can also have synergistic effects, as reported previously. Wang et al. found that different *Fusarium* species enhanced *P. sojae* infection when co-inoculated on soybean. Co-infection with *Fusarium* promotes the loss of *P. sojae* resistance in soybeans [45]. Whitelaw-Weckert et al. found that co-inoculation of *Ilyonectria* and *Botryosphaeriaceae* isolates led to increased disease severity compared to monoculture inoculations of *Ilyonectria* isolates [46]. In our study, compared to single inoculation, co-infection with Dt and Cf in tea leaves resulted in larger lesions. Interestingly, an aggravation of the condition was observed in dual inoculation of Dt and Cf on tea leaves, but not in Dt and Cc. According to previous studies and pathogenicity tests in this study, we found that Cc is more virulent than Dt and Cf, and low concentrations of spore suspensions of Dt and Cf do not cause disease spot occurrence on tea leaves [13,16]. Co-inoculation with different concentrations of spore suspensions was performed to explore the effect of Cc, Cf, and Dt during the infection. At the low concentration (Cf cannot cause tea plant disease in this concentration range), the dual inoculation of Dt and Cf caused lesions in tea leaves and significantly enhanced the infection capacity compared with single inoculation. We found that the interaction of multiple pathogens may change the severity of the disease. However, Kaur et al. pre-inoculation with *Hyaloperonospora parasitica* (an asymptomatic isolate) reduces the incubation period and increases susceptibility of a *Brassica* to white rust disease [47]. This is different from our results. Our results indicated that the single inoculation and dual inoculation related to virulent Cc had similar results. We considered that the temporal order of host infection may play a role in host–multi-pathogen interaction. Le May et al. revealed that the severity of disease was different between inoculation of two pathogens simultaneously and sequentially [44].

For a long time, research on host–pathogen interaction was in a one-to-one mode where a single pathogen colonizes a host. Disease control measures were also focused on dealing with specific pathogens. However, addressing diseases caused by multiple species requires new measures. According to our results, we believe that the main cause of anthracnose might be the concerted action of a variety of fungi. In the future, the interaction mechanisms of Dt and *Colletotrichum* and their relationship with tea plants deserve more attention.

## 4. Materials and Methods

### 4.1. Collection and Isolation

Tea leaves were collected from *Camellia sinensis* cv. *Longjing43* (LJ43) in three different tea gardens of Zhejiang Province, China. Among each tea garden, more than three tea plant individuals were sampled with diseased tea leaves fitting the symptoms as P1, P2, and P3. In addition, three diseased tea leaves with equivalent leaf positions were sampled randomly in each individual. The sampling period was from June to August continuously for two years. Diseased samples were identified and collected from leaves with visible anthracnose symptoms, which are sub-round or irregular scabs, scattered small black sporophores, water-soaked lesions, and the absence of thin ring spots [48]. The isolates separated from leaves were obtained by a single spore isolation technique described by Cai et al. [49]. In addition, the isolates separated from a healthy part of tea diseased leaves were collected according to a protocol from Kjer et al. [50]. Briefly, the fungi were collected after washing diseased leaves with tap water, alcohol (70%) for 1 min, sodium hypochlorite (2.5%) for 1 min, and then washed in sterile distilled water. To check the efficacy of this method of surface sterilization, 100 µL of the last wash water was incubated on PDA plates.

### 4.2. Morphological Characterization

Mycelial discs were taken from the margin of 5-day-old PDA cultures and incubated on PDA plates in the dark at 25 °C. For growth rate measurement, each isolate was measured in at least three independent experiments. The colony diameter was measured daily beginning 3 days after inoculation and used for calculating the growth rate. The growth rate can be calculated as the average of mean daily growth (mm/d). The colony and conidial characteristics were determined via methods described by Cai et al. [49]. Appressoria was produced and measured using a slide culture technique and induced on a synthetic nutrient-poor agar (SNA) medium.

### 4.3. DNA Extraction and PCR Amplification

Genomic DNA was extracted from mycelium cultured on PDA for 7 days. Total genomic DNA, which was used as the template for PCR amplification, was extracted using an Ezup Column Fungi Genomic DNA Purification Kit [Sangon Biotech (Shanghai) Company Limited, Shanghai, China] and stored at −20 °C. Four partial loci were used for *Colletotrichum* identification, including ribosomal internal transcribed spacer (ITS), glyceraldehyde 3-phosphate dehydrogenase (GAPDH), beta-tubulin (TUB2), and calmodulin (CAL) genes. Other partial loci, such as ITS, the partial 28S nrDNA (LSU), DNA-directed RNA polymerase II second largest subunit (RPB2), and translation elongation factor 1-alpha (EF1) were used to identify *Diaporthales*. The primers used in this study are listed in Appendix A, while the polymerase chain reaction (PCR) was performed as follows: 4 min at 94 °C; followed by 35 cycles of 30 s at 94 °C; 50 s at 57 °C (LSU) or 54 °C (EF1) or 55 °C (RPB2); 30 s at 72 °C, and 10 min at 72 °C. The PCR conditions for ITS, GAPDH, TUB2, and CAL were as described by Lu et al. [16]. PCR purification and DNA sequencing were performed by Shanghai Huagene Biotech Company Limited, Shanghai, China.

### 4.4. Phylogenetic Analysis

The accession numbers of the sequences were obtained from NCBI GenBank based on BLAST searches and published literature and are listed in Appendix A. Sequences from forward and reverse primers were edited with SeqMan and DNAStar. We used Bayesian inference (BI) to construct the phylogenies according to multilocus sequence analysis. MrBayes v. 3.2. MrModel test v. 2.3 was used to select the best-fit models of nucleotide substitution, then assembling and modulating the dataset by MAFFT v.7 and MEGA v. 6.0, respectively. All gaps were regarded as missing data. Markov chain Monte Carlo (MCMC) sampling was used to reconstruct the phylogenies in MrBayes v.3.2. Analyses of 6 MCMC chains based on the full dataset were run for 2 × 10^7^ generations and sampled every 100 generations [13,16].

### 4.5. Pathogenicity Tests of Discula Theae-Sinensis

First, the healthy, non-wounded the third tea leaves were washed with tap water and then sterilized with 1% sodium hypochlorite for 2 min, flushed with sterilized water 3 times, and air dried. Afterward, sterile needles were used to puncture the leaves to make wounded spots. Subsequently, 5 mm mycelial discs from a 5-day-old culture were placed over every wound of tea leaves. The inoculated leaves were cultured in an incubator at 25 °C with a 12/12 h (day/night) photoperiod. The conditions of inoculated leaves were recorded at 5 and 7 days post-inoculation. In this experiment, three replicates were included for each treatment. Sterile-distilled PDA discs with no mycelia were used as controls. Finally, Koch’s postulates were employed to investigate each strain. Test of each isolate was performed in three biological replicates.

### 4.6. Dual Inoculation Assays of Dt and Colletotrichum

The plate confrontation culture method was used to assess the mutual effect between Dt and *Colletotrichum* spp. in vitro. The PDA plates were cultured in an incubator at 25 °C for 5 days in the dark. And then, observed the mutual effect between strains. The formula of inhibition ratio of growth of fungi was calculated described by López-González et al. was followed with minor revisions [51].
Inhibition ratio (%) = [(R_1_ − R_2_)/R_1_] × 100 (1)
where R_1_ indicates the radial growth of the fungi colony in control plates and R_2_ indicates the radial growth of the fungi colony (in the direction of the other fungus).

Dual assays in vivo were used to explore the relationship between Dt and *Colletotrichum* spp. during the procession of infection. The method of leaf preparation and inoculation condition was the same as the pathogenicity test. The inoculation experiments were grouped into 8 treatments: Dts, Dtd, Cc, Cf, Dts + Cc, Dtd + Cc, Dts + Cf, and Dtd + Cf. The labels and corresponding treatment conditions are shown in Table 1. The concentration of spore suspension was 10^6^ spores/mL in these experiments. Distilled water was used as the control.

**Table 1 plants-12-03427-t001:** Labels and corresponding treatment conditions in dual inoculation.

Labels	Treatment Conditions
Dts	single inoculation of spore suspension of Dt
Dtd	single inoculation of mycelial discs of Dt
Cc	single inoculation of spore suspension of Cc
Cf	single inoculation of spore suspension of Cf
Dts + Cc	dual inoculation with a spore suspension of Dt and Cc
Dtd + Cc	dual inoculation with mycelial discs of Dt and a spore suspension of Cc
Dts + Cf	dual inoculation with spore suspension Dt and Cf
Dtd + Cf	dual inoculation with mycelial discs of Dt and spore suspension of Cf

### 4.7. Impact of Spore Suspension Concentration on Combined Dt and Cf Infection Capacity

To explore the interaction of Dt and Cf at various concentrations in a spore suspension, inocula were prepared by mixing spore suspensions of Dt and Cf in equal amounts. The concentration of the spore suspension was 10^5^ spores/mL, 10^6^ spores/mL, and 10^7^ spores/mL. Six single inoculations and 9 combinations were obtained. The labels and corresponding treatment conditions are shown in Table 2.

**Table 2 plants-12-03427-t002:** Labels and corresponding treatment conditions in this sector.

	Labels		Treatment Conditions
Single inoculation		Dt5	10^5^ spores/mL spore suspension of Dt
		Dt6	10^6^ spores/mL spore suspension of Dt
		Dt7	10^7^ spores/mL spore suspension of Dt
		Cf5	10^5^ spores/mL spore suspension of Cf
		Cf6	10^6^ spores/mL spore suspension of Cf
		Cf7	10^7^ spores/mL spore suspension of Cf
Dual inoculation	F5 group	Dt5Cf5	10^5^ spores/mL Dt and 10^5^ spores/mL Cf
		Dt6Cf5	10^6^ spores/mL Dt and 10^5^ spores/mL Cf
		Dt7Cf5	10^7^ spores/mL Dt and 10^5^ spores/mL Cf
	F6 group	Dt5Cf6	10^5^ spores/mL Dt and 10^6^ spores/mL Cf
		Dt6Cf6	10^6^ spores/mL Dt and 10^6^ spores/mL Cf
		Dt7Cf6	10^7^ spores/mL Dt and 10^6^ spores/mL Cf
	F7 group	Dt5Cf7	10^5^ spores/mL Dt and 10^7^ spores/mL Cf
		Dt6Cf7	10^6^ spores/mL Dt and 10^7^ spores/mL Cf
		Dt7Cf7	10^7^ spores/mL Dt and 10^7^ spores/mL Cf

### 4.8. Statistical Analysis and Microscopic Observation

IBM SPSS Statistics software version 19 was used to analyze variance (ANOVA) of the experimental data. The mean values were compared by the least significant difference (LSD) method, and differences were considered significant at *p* < 0.05. All data are represented as the average ± standard error.

An OLYMPUS stereo microscope was used to observe acervuli on diseased tea leaves from the field. A Nikon 80i microscope equipped with a differential interference contrast (DIC) objective was used to visualize spores on diseased tea leaves. Lesion observation after dual inoculation was performed using a field-depth microscope (KEYENCE). Conidial characteristics were recorded by ZEISS Axio Vert. A1.

## 5. Conclusions

In this study, we delved into the dominant species responsible for anthracnose in tea plants and conducted an extensive exploration of the diversity within Dt and *Colletotrichum* associated with *Camellia sinensis*. Our findings revealed that Dt is the predominant species in tea leaves, serving as the primary causative agent of tea plant anthracnose. Furthermore, we identified a total of eight known *Colletotrichum* species with the noteworthy inclusion of *C. tropicicola* and *C. gigasporum*, which were reported for the first time in tea plants. Our investigation also highlighted the advantages of co-inoculation with Dt and *C. fructicola*, particularly at low concentrations, as it demonstrated superior outcomes compared to single inoculations. This observation suggests that the interaction between Dt and *Colletotrichum* may play a pivotal role in the development of anthracnose in tea plants. In conclusion, our research sheds new light on the multifaceted nature of anthracnose in tea plants. Rather than a single pathogenic agent, our findings suggest that anthracnose may result from the collaborative efforts of various fungal species. As we move forward, it is imperative to delve deeper into the mechanisms of interaction between Dt and *Colletotrichum* and their intricate relationship with tea plants. This knowledge will be essential for the development of effective strategies for anthracnose control and the continued prosperity of the tea industry.

## Figures and Tables

**Figure 1 plants-12-03427-f001:**
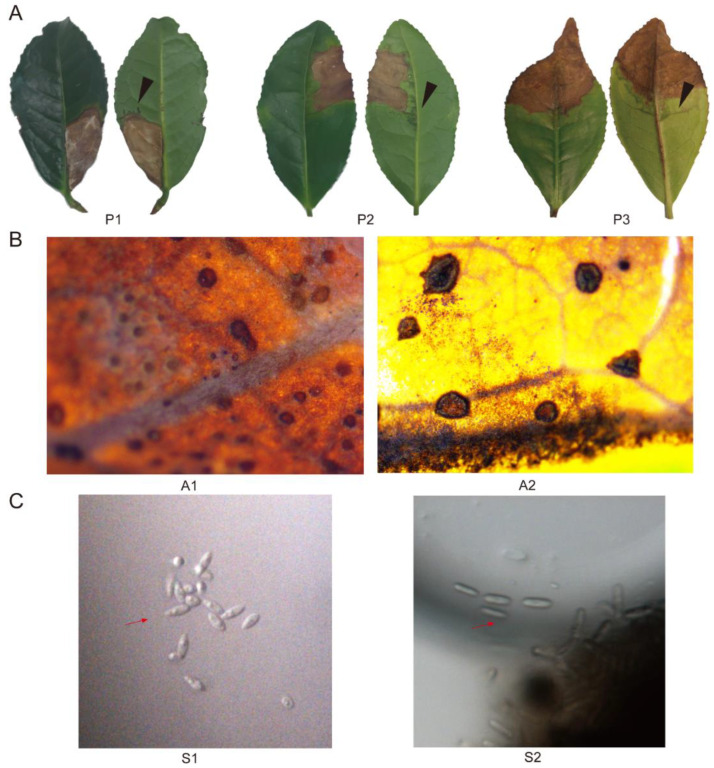
Phenotypic characterization of diseased tea leaves. (**A**) Morphology of tea leaves. Black triangles indicate water-soaked lesions. (**B**) Shape of acervuli on tea leaves. (**C**) Spores isolated from lesions. Red arrows indicate spores.

**Figure 2 plants-12-03427-f002:**
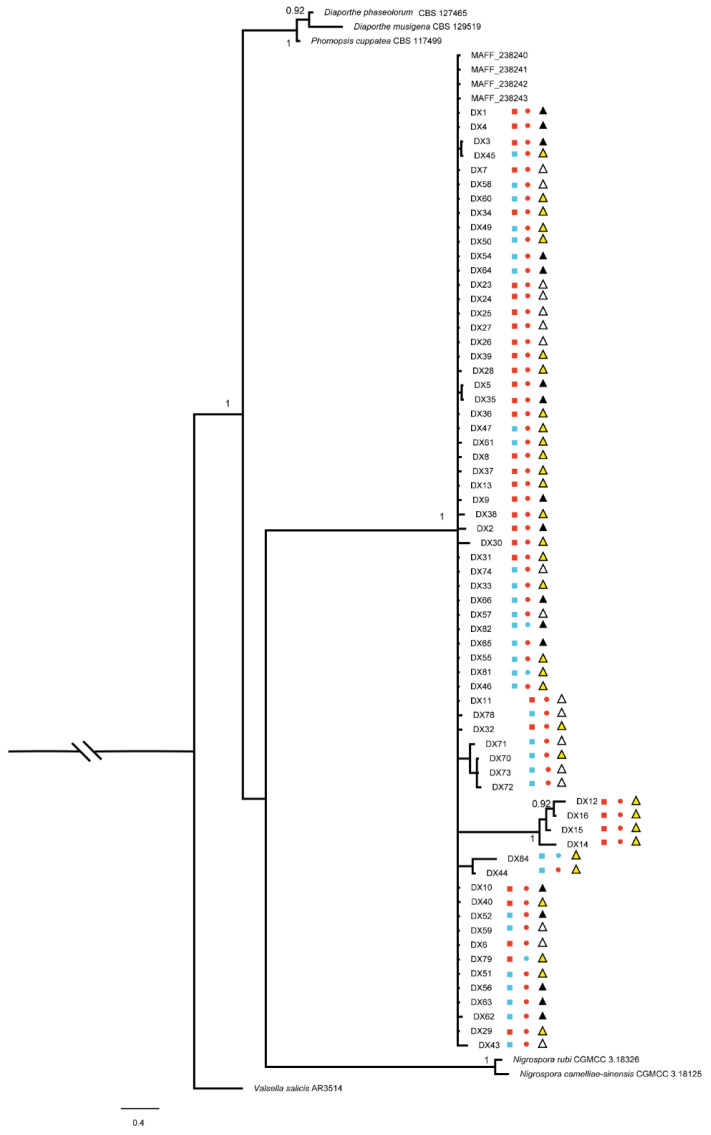
Phylogenetic tree generated by Bayesian analysis based on a combined 4-gene dataset (ITS, EF1, RPB2, and LSU). Bayesian posterior probabilities above 0.90 are shown at each node. *Valsella salicis* AR3514 was used as the outgroup. The scale bar indicates 0.4 expected changes per site. The branches crossed by diagonal lines are shortened by 50%. Red squares indicate strains isolated in 2019; blue squares indicate strains isolated in 2020; red circles indicate isolates collected from diseased part of tea diseased leaves; blue circles indicate isolates collected from healthy part of tea diseased leaves; white triangles indicate strains isolated from P1; yellow triangles indicate strains isolated from P2; black triangles indicate strains isolated from P3.

**Figure 3 plants-12-03427-f003:**
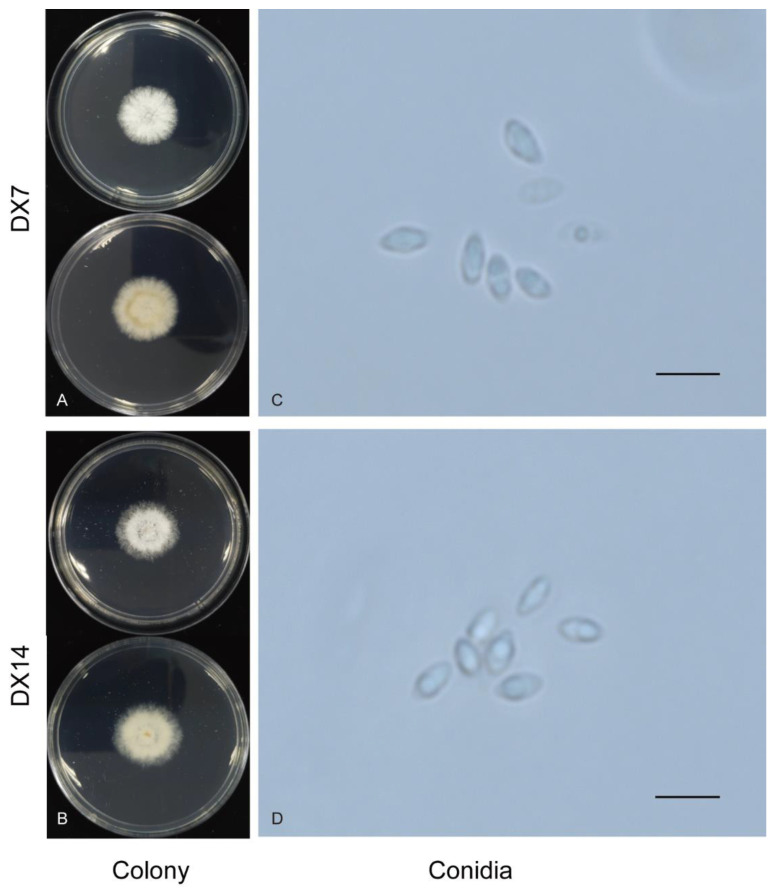
Morphological characteristics of Dt. (**A**,**B**) colonies of DX7 and DX14. (**C**,**D**) conidia of DX7 and DX14, Scale bar = 80 µm.

**Figure 4 plants-12-03427-f004:**
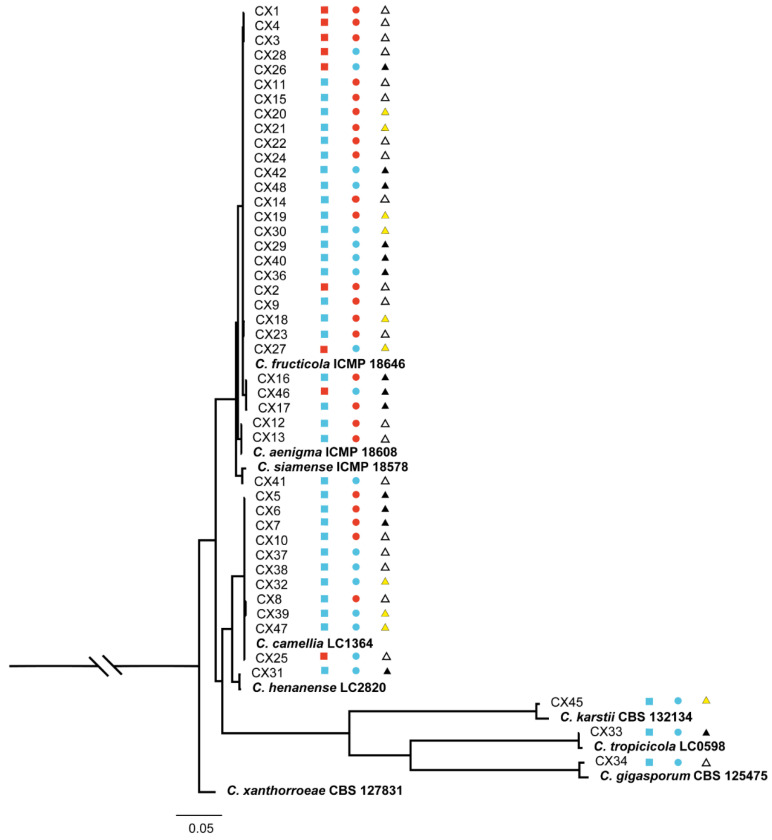
Phylogenetic tree generated via Bayesian analysis based on a combined 4-gene dataset (ITS, CAL, TUB2, and GAPDH). Bayesian posterior probabilities above 0.95 are shown at each node. *C. xanthorrhoeae* CBS 127831 was used as the outgroup. The scale bar indicates 0.05 expected changes per site. The ex-type strains are emphasized in bold. The branches crossed by diagonal lines are shortened by 50%. Red squares indicate strains isolated in 2019; blue squares indicate strains isolated in 2020; red circles indicate isolates collected from diseased part of tea diseased leaves; blue circles indicate isolates collected from healthy part of tea diseased leaves; white triangles indicate strains isolated from P1; yellow triangles indicate strains isolated from P2; black triangles indicate strains isolated from P3.

**Figure 5 plants-12-03427-f005:**
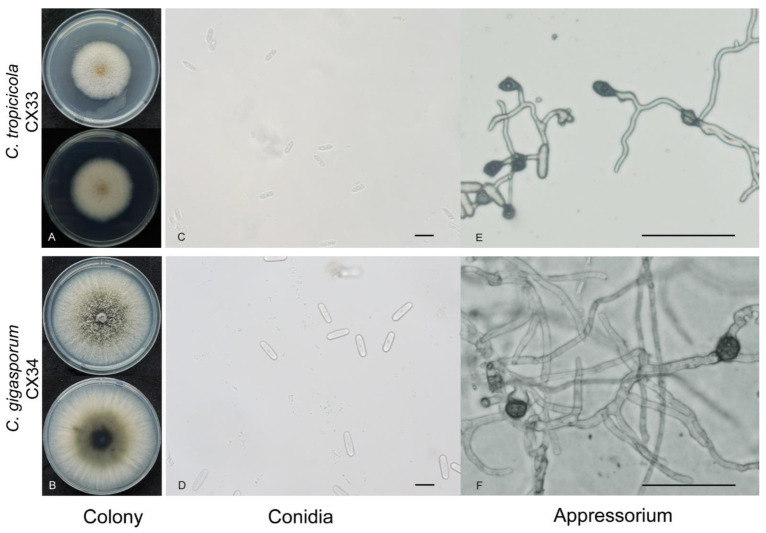
Morphological characteristics of *C. tropicicola* and *C. gigasporum*. (**A**,**B**) colonies of *C. tropicicola* (CX33) and *C. gigasporum* (CX34). (**C**,**D**) conidia of *C. tropicicola* (CX33) and *C. gigasporum* (CX34), Scale bar = 20 µm. (**E**,**F**) Appressoria of *C. tropicicola* (CX33) and *C. gigasporum* (CX34), Scale bar = 50 µm.

**Figure 6 plants-12-03427-f006:**
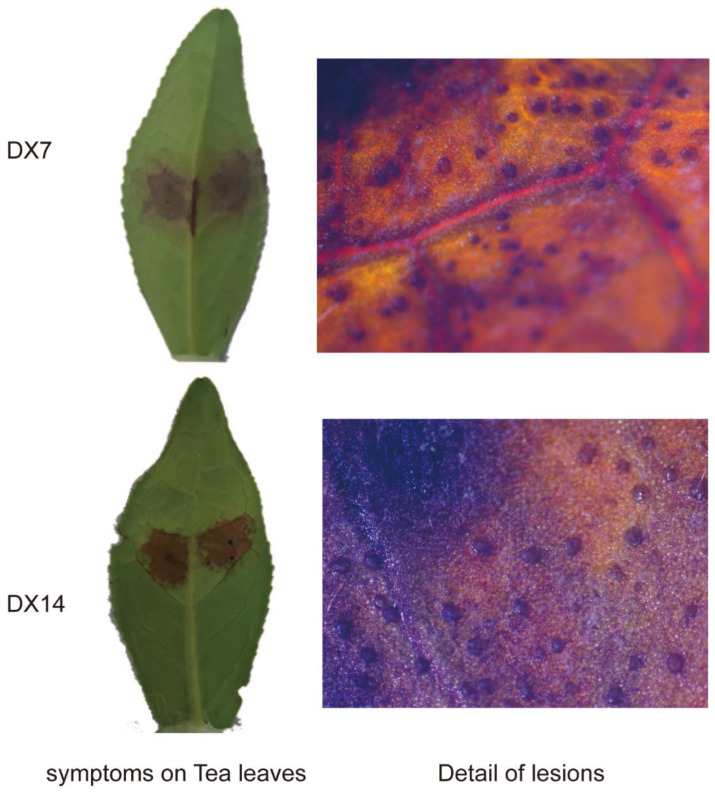
Pathogenicity tests of Dt on tea plant leaves after inoculation.

**Figure 7 plants-12-03427-f007:**
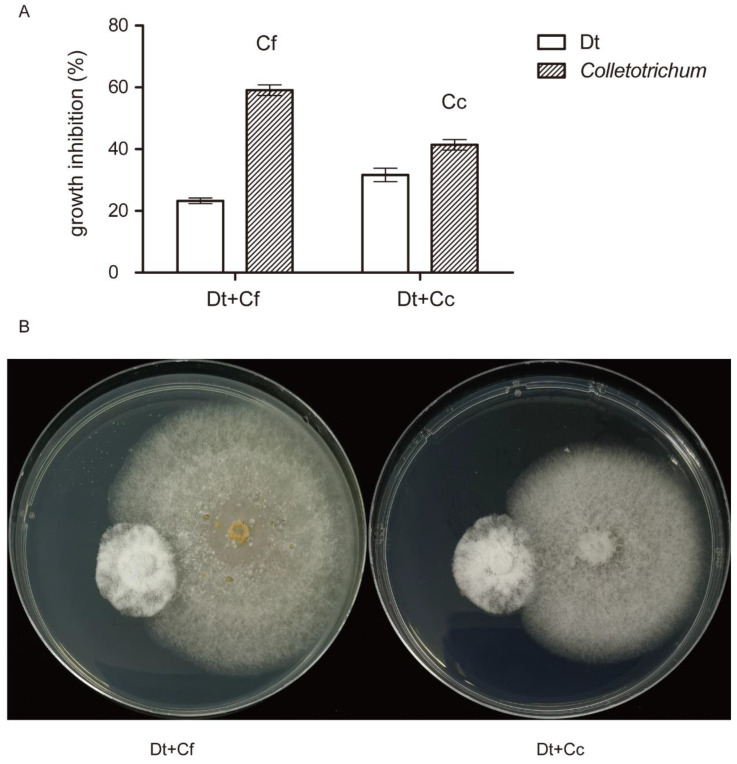
Dual inoculation in vitro at 5 days. (**A**) Growth inhibition of Dt and *Colletotrichum.* (**B**) Plate confrontation culture between Dt (**left**) and *Colletotrichum* (**right**).

**Figure 8 plants-12-03427-f008:**
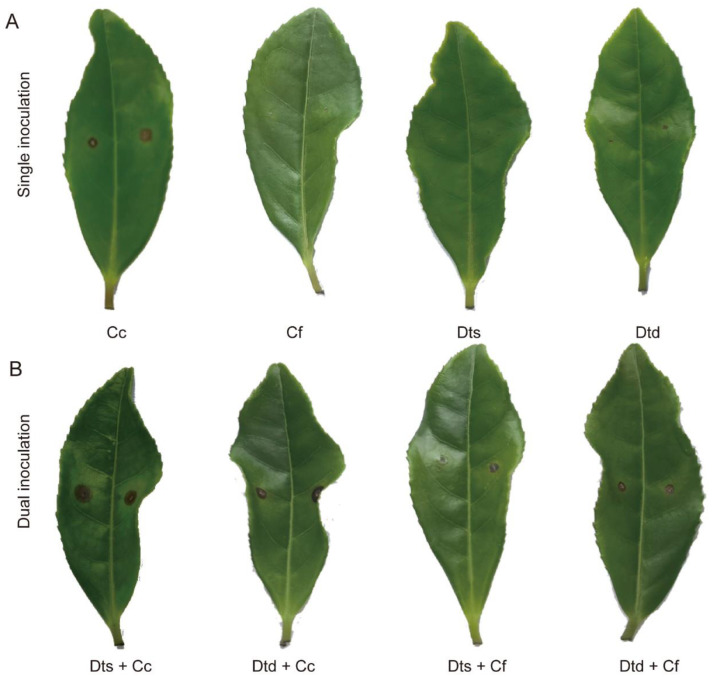
Dual inoculation in vivo. (**A**) Results of single inoculation. (**B**) Results of dual inoculation. The meaning of labels is shown in Table 1.

**Figure 9 plants-12-03427-f009:**
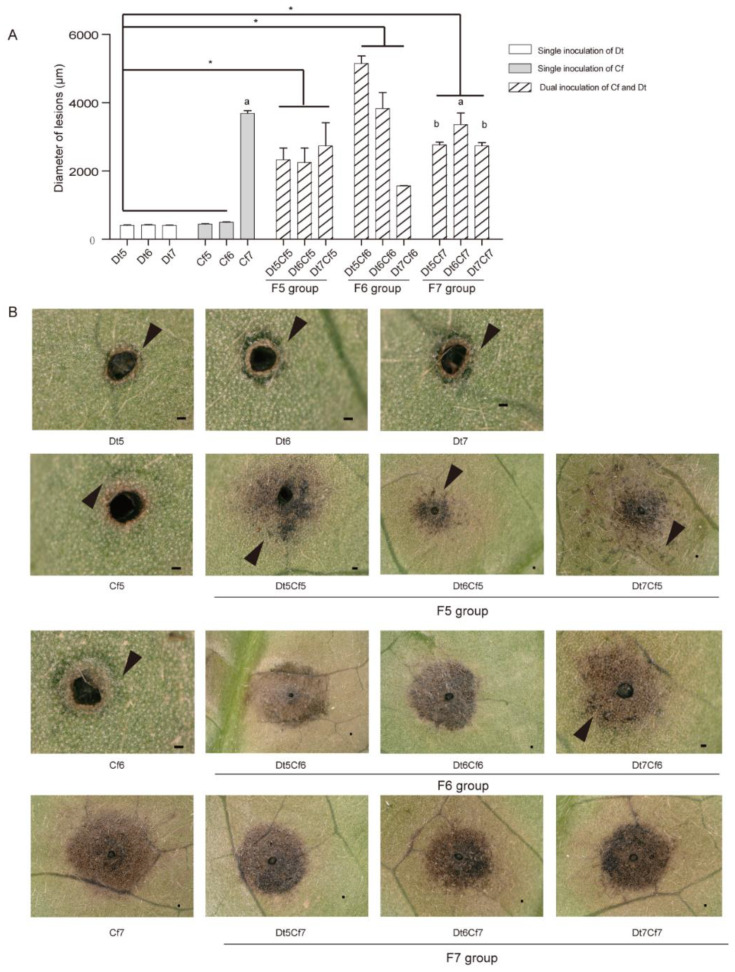
The interaction of Cf and Dt on tea plant infection. (**A**) Bar chart of the diameter of lesions of different inoculation treatments. Asterisks indicate a significant difference between single inoculation of Dt, Cf5, Cf6, and Dual inoculation according to one-way ANOVA (*p* value < 0.05). letters represent differences between Cf7 and dual inoculations of F7 group. The Error bars indicate standard error (SE). (**B**) Images of lesions after different inoculation treatments. Black triangles indicate water-soaked lesions, scale bar = 100 µm. The meaning of labels is shown in Table 2.

## Data Availability

Data are available upon request from the corresponding author.

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
