# Peer review of "Characteristics and Pathogenicity of Discula theae-sinensis Isolated from Tea Plant (Camellia sinensis) and Interaction with Colletotrichum spp."

_plants, 2023, doi:10.3390/plants12193427_

Round 1

Reviewer 1 Report

The manuscript identified the pathogenic fungi causing tea anthracnose via multilocus phylogenetic and morphological analysis as well as pathogenicity tests. It is a lot of work to isolate multiple fungal strains in the manuscript. However, i have some concerns showed below that should be elaborate:

For morphological identification of Diaporthe species, α- and β-conidia, stroma were the key points that should be described in the manuscript. Identically, the conidia and appressorium of Colletotrichum species also should be compared. Importantly, reference regarding to each specific species classification should be cited for morphological identification.

For pathogenicity tests, each identified strain needs to contain a representative strain, including the suspected endophyte.

In Fig.5, the front and reverse sides of the plates cultivated at 15d need to be reversed.

In Fig.9, if the authors want to compare the virulence between different strains, it is best to count the size of the lesions.

Author Response

Dear editor/reviewer,

We express our gratitude for your valuable comments. We have revised the manuscript, and would like to re-submit it for your consideration. All comments raised by the reviewers were addressed, and the amendments are highlighted in yellow in the revised manuscript. Also, the revised manuscript has been re-edited by a native English speaker from an editing company. Point by point response to the reviewers’ comments are listed in details below.

Your sincerely,

Lu Qinhua

Replies to Reviewer 1:

Comments to the author

1) In morphological identification section, something important, such as characteristics of α- and β-conidia, stroma and appressoria, should be be described in the manuscript.

Response: Thank you for your constructive comments. According to the published articles, the α-conidia and stroma can be observed in the Diaporthales species, while the β-conidia only can be observed occasionally. Based on our several previous experiments and this study, we only spotted the α-conidia of Diaporthales species. Combined with the third query of the Reviewer 4 and the fifth query of Reviewer 5, and our whole study is focused on the Discula theae-sinensis, to scrupulously operate this study, we decided to delete the other species in Diaporthales besides Discula theae-sinensis in the manuscript. Moreover, researchers prefer use molecular and morphological identification together to pinpoint the fungi. In this study, we used ITS, LSU, RPB2, EF1 and morphological observation (α-conidia) to define Discula theae-sinensis. In addition, previous study also cannot spot the β-conidia and stroma of the Discula theae-sinensis. (Moriwaki, J. & Sato, T., 2009). According to the Colletotrichum species, condia and appressoria can be observed, so we have re-operated this experiment and added the results.

Reference: Moriwaki, J.; Sato, T. A new combination for the causal agent of tea anthracnose: Discula theae-sinensis (I. Miyake) Moriwaki & Toy. Sato, comb. nov. J. Gen. Plant Pathol. 2009, 75, 359-361.

2) For pathogenicity tests, each identified strain needs to contain a representative strain, including the suspected endophyte.

Response: Thank you for your constructive comments. Based on phylogenetic analysis, all separated strains can be divided into two major clusters. Among the two clusters, we chose representative strain of each cluster randomly (DX7 and DX14 respectively) for pathogenicity test. In addition, DX7 and DX14 have morphological identification results. Combined with the first query of the Reviewer 2, the expression of “endophyte” in this manuscript may be not accurate. The former definition of “endophyte” in this study is separating the fungi from healthy part of diseased leaves. According to your advice and other reviewers’ advice, we deleted the expression of “endophyte” in the revised version, instead of it, we used the direct expression of “fungi from healthy part of diseased leaves”.

3) In Fig.5, the front and reverse sides of the plates cultivated at 15d need to be reversed.

Response: Combined with the third query of the Reviewer 4 and the fifth query of Reviewer 5, we deleted the other species in Diaporthales besides Discula theae-sinensis in the manuscript. So, the figure 5 is also deleted in this part.

4) In Fig.9, if the authors want to compare the virulence between different strains, it is best to count the size of the lesions.

Response: Thank you for your constructive comments. We agreed with this view completely that quantitative data of lesions size was the optimal means to compare the virulence. According to our scaled data, the lesions diameters of D1 and D2 are 0.5 times and 3.9 times than Dt (0.4 ± 0.04 cm) in mature leaves, respectively. Because we focus on Discula theae-sinensis in this manuscript now, the results of D1 and D2 are not listed in the present text. We may operate more experiments to study the characteristics of D1 and D2 in the future. In revised version, we demonstrated pathogenicity of DX7 and DX14. Both of these two strains were Dt and have similar lesions sizes.

Reviewer 2 Report

The manuscript “Characteristics and Pathogenicity of Discula theae-sinensis Causing Camellia sinensis Anthracnose, and Synergistic Effect with Colletotrichum spp.” describe the fungi identified in anthracnose lesions in Camellia sinensis. The authors identified fungi belonging to the Colletotrichum genus as well as fungi from Diaporthales order, such as Discula theae-sinensis. Besides that, the authors tried to show the existence of a synergetic effect in the presence of fungus from Diaporthales order and Colletotrichum.

However, the manuscript has too many scientific issues to be accepted in the present form. Namely, it is not clear the results obtained from the Koch’s postulates tests, due to that it is not clear which isolates were endophytic, saprobes and pathogens; wounding the leaves is mandatory for the development of the disease? Subsequently, the interaction effect of the presence of Diaporthales and Colletotrichum does not provide sufficient empirical evidence.

Beyond that, the literature review is too old. It lacks the last updates on Colletotrichum taxonomy.

Other issues

Line 68 – The authors should update the information about Colletotrichum taxonomy!

Line 74-76 - The sentence: “Cc is more virulent than Cf on tea plants, which may be caused by secondary metabolites, such as catechins and caffeine [21],” does not make sense, the authors should explained what they want to say.

Figure 1. – Fig 1B and 1C - Photographs do not have quality; besides that, the authors do not referred if the acervuli and the spores observed were related with the type of symptoms observed.

Line 138-139 – The morphological characteristics depend in the inoculation medium as well as the incubation conditions. All the parameters that explain the experimental conditions should be report, with as much detailed as possible. The authors should include all those informations.

Line 143 – 160 – A table or a Box and Whisker Plots will help the readers to visualise the results.

Line 155-156 - D2(DX22) the presented value of 2.41 mm per day are not in agreement with the value presented in the chart (figure 3).

Figure 4, 5 and 6 should be merged in one figure. The photographs of the fungal growth on PDA should show both sides of the plate. The photograph by 15 days is unnecessary, several isolates had fulfilled all the plate. The scale bar is only related with a part of the figure, so the caption must be divided. Moreover, the photomicrographs do not have quality enough for being presented in a scientific manuscript.

Figure 8 – Why a special focus on putatively endophytic isolates?

Figure 9 – The authors know if there are any effect on the development of the disease when using detached leaves in comparison to plant inoculations?

Line 315-321 – Difficult to understand and it seems irrelevant.

Author Response

Dear editor/reviewer,

We express our gratitude for your valuable comments. We have revised the manuscript, and would like to re-submit it for your consideration. All comments raised by the reviewers were addressed, and the amendments are highlighted in yellow in the revised manuscript. Also, the revised manuscript has been re-edited by a native English speaker from an editing company. Point by point response to the reviewers’ comments are listed in details below.

Your sincerely,

Lu Qinhua

Replies to Reviewer 2:

Comments to the author

1) It is not clear the results obtained from the Koch’s postulates tests, due to that it is not clear which isolates were endophytic, saprobes and pathogens.

Response: 

Thank you for your constructive comments. We have used Discula theae-sinensis to infect the tea leaves following the Koch’s postulates tests instruction during the study. Combined with the third query of the Reviewer 4 and and the fifth query of Reviewer 5, and our whole study is focused on the Discula theae-sinensis, to scrupulously operate this study, we decided to delete the other species in Diaporthales besides Discula theae-sinensis in the manuscript. In addition, we added the results of the Koch’s postulates tests including the morphological symptoms and lesion details. Based on the Koch’s postulates tests, the laboratory (figure 6) and field results (figure 1, A1) are consistent. We collected isolates in this study from diseased part and healthy part of diseased leaves. The expression of “endophyte” in this manuscript may be not accurate. The former definition of “endophyte” in this study is separating the fungi from healthy part of diseased leaves. According to your advice and other reviewers’ advice, we deleted the expression of “endophyte” in the revised version, instead of it, we used the direct expression of “fungi from healthy part of diseased leaves”.

2) Wounding the leaves is mandatory for the development of the disease?

Response: This is a constructive advice and always raised by reviewers and readers. To enhance the stability of such experiments, wounded inoculation is a highly accepted and recommended method for studying the fungi characteristic in tea area in vivo. Most of the previous studies chose wounded inoculation to explore the interaction between tea plant and fungi. For example, Wang et. al. [1] adopted wounded inoculation to compare transcriptomic changes in susceptible tea cultivar Longjing43 and the resistant cultivar Zhongcha108. Zhou et. al. reported a mycovirus that modulates endophytic and phytopathogenic fungal traits [2]. To verify the reliability of this study, we chose the most admissive research method.

Reference:

[1] Wang Y, Hao X, Lu Q, Wang L, Qian W, Li N, Ding C, Wang X, Yang Y. Transcriptional analysis and histochemistry reveal that hypersensitive cell death and H2O2 have crucial roles in the resistance of tea plant (Camellia sinensis (L.) O. Kuntze) to anthracnose. Hortic Res. 2018, 5, 18

[2] Zhou, L., Li, X., Kotta-Loizou, I. A mycovirus modulates the endophytic and pathogenic traits of a plant associated fungus. ISME. 2021, 15, 1893-1906.

3) It lacks the last updates on Colletotrichum And Line 68 – The authors should update the information about Colletotrichumtaxonomy!

Response: According to the reviewer’s constructive comment, we have updated information about Colletotrichum taxonomy and modified the manuscript. The latest article about Colletotrichum species identification published in 2023 was added into the introduction [1]. We have altered the text into “Through sampling in multiple regions, 21 species and 1 indistinguishable strain of Colletotrichum have been reported to infect tea plants in China”.

Reference:

[1] Peng XJ, Wang QC, Zhang SK, Guo K, Zhou XD 2023 – Colletotrichum species associated with Camellia anthracnose in China. Mycosphere 14(2), 130–157,

4) Line 74-76 - The sentence: “Cc is more virulent than Cf on tea plants, which may be caused by secondary metabolites, such as catechins and caffeine [21],” does not make sense, the authors should explain what they want to say.

Response: Thank you for your suggestion. Our original purpose of this sentence want to express two meanings. Firstly, Cc is more virulent than Cf on tea plants; secondly, secondary metabolites of tea plant may have an effect on the virulence of pathogen. Because of our poor expression, it becomes ambiguity and less logical. So we have revised the text into two sentences.

5) Fig 1B and 1C - Photographs do not have quality; besides that, the authors do not refer if the acervuli and the spores observed were related with the type of symptoms observed.

Response: Thank you for your constructive comments. Fig 1B and 1C was taken while collecting samples, it's hard to get new pictures. So we adjusted the scale of the picture and replaced old one. On account of geographical distribution and different disease courses, the morphology of disease spots were different. It is difficult to determine the type of disease only by observation. So we want to figure out the relationship between acervuli and the type of symptoms is the original idea of this study. We found the correlation between Cf and type A2 acervuli was high. According to experimental record, type A2 acervuli was found in P1, but appears only once during the experiment. In addition, we used Discula theae-sinensis to infect the tea leaves, the both P1 and P2 symptoms appear on diseased leaves.

6) Line 138-139 – The morphological characteristics depend in the inoculation medium as well as the incubation conditions. All the parameters that explain the experimental conditions should be report, with as much detailed as possible. The authors should include all information.

Response: Thank you for your constructive comments. We have added parameters, such as culture medium, temperature, light condition in Section 2.2. Identification and Characterization of Discula theae-sinensis. Fungi were incubated on PDA plates in the dark at 25 oC.

7) Line 143 - 160 – A table or a Box and Whisker Plots will help the readers to visualise the results.

Response: Thank you for your constructive comments. In the previous version, we listed the details of multiple strains, and it is hard to understand for the readers. However, we have deleted the redundant content in the present version, only two strains existed in the Discula theae-sinensis and Colletotrichum species. So we decided to retain them in text.

8) Line 155-156 - D2(DX22) the presented value of 2.41 mm per day are not in agreement with the value presented in the chart (figure 3).

Response: Thank you for your comments. This is our mistake. After rechecking the data, the correct number is 4.32 mm. However, we deleted the information about D2 in revised version. According to your suggestion, we have checked the other data to make sure they are right.

9) Figure 4, 5 and 6 should be merged in one figure. The photographs of the fungal growth on PDA should show both sides of the plate. The photograph by 15 days is unnecessary, several isolates had fulfilled all the plate. The scale bar is only related with a part of the figure, so the caption must be divided. Moreover, the photomicrographs do not have quality enough for being presented in a scientific manuscript.

Response: Thank you for your comments. According to your suggestion, we have deleted the photograph of 15 days, and both sides of the plate have been arranged in the manuscript. The Diaporthales conidia is rather small, we have replaced the picture and adjusted the pixels as much as possible. In the revised manuscript, we change the scale of photos to make them clearer than before. Combined with the third query of the Reviewer 4 and the fifth query of Reviewer 5, we deleted the former Figure 5 and Figure 6 in the latest version.

10) Figure 8 – Why a special focus on putatively endophytic isolates?

Response: This is a constructive comment. In this study, we collected 45 isolates belonged to Colletotrichum. All these isolates belonged to 8 known species, in which C. tropicicola and C. gigasporum were first reported in tea plants. Other 6 known species collected from tea plants have been described in previous studies. So we describe morphological characteristics of C. tropicicola and C. gigasporum in details. The fundamental reason is the two isolates are new reported in tea plants, not because they are endophytes.

11) Line 315-321 – Difficult to understand and it seems irrelevant.

Response: Thank you for your constructive comments. In this part, we wanted to explore the conditions of the occurrence of water-soaked lesions. In the field, water-soaked lesions are common phenomenon of tea anthracnose. Based on results of dual inoculation, we found water-soaked lesions may be influenced by virulence of fungi. Thus, we wrote this paragraph. In revised manuscript, we added the contexts about meaning of water-soaked lesions and modified the ambiguous sentences.

Reviewer 3 Report

Lines 16-17
"Identifying the specific pathogenic fungi causing tea anthracnose is an essential control measure to mitigate this disease"

Please, explain. Are the different pathogenic fungal species eliminated from the tea roots in different, species-specific ways? If not, why species-lvl identification of pathogens is so important? Please, clarify it in the Introduction section.

Please, involve potential implementation of the presented results in tea cultivation and plant protection.

Line 320
"To explore the composition and function watery lesions may be a meaningful research in the future. "

This sentence should be placed in the Discussion section and then extended.

Methods
Please clarify, how many plant individuals (not leaves) were studied? How much the results differed between different plant individuals within and/or between three study sites?

Lines 430-431
"Tea leaves were collected from Camellia sinensis cv. Longjing43 (LJ43) in three different tea gardens of Zhejiang Province, China."

Are only three sites enough to study the participation of each pathogenic fungal species in tea leaves in tea plantations in general?  Please, explain it using well-published studies.

Please, modify the title, as follows:
Characteristics of pathogenic fungi responsible for anthracnose of Camellia sinensis leaves in three tea gardens located in Zhejiang Province in China.

Supplementary files
All references should be added to the Reference list.

Conclusions
Line 425
"The main cause of anthracnose might be the concerted action of a variety of fungi."

Please, expand this sentence and place it as the main conclusion of the study in the Abstract and Conclusion sections.

none

Author Response

Dear editor/reviewer,

We express our gratitude for your valuable comments. We have revised the manuscript, and would like to re-submit it for your consideration. All comments raised by the reviewers were addressed, and the amendments are highlighted in yellow in the revised manuscript. Also, the revised manuscript has been re-edited by a native English speaker from an editing company. Point by point response to the reviewers’ comments are listed in details below.

Your sincerely,

Lu Qinhua

Replies to Reviewer 3:

1) "Identifying the specific pathogenic fungi causing tea anthracnose is an essential control measure to mitigate this disease"

Please, explain. Are the different pathogenic fungal species eliminated from the tea roots in different, species-specific ways? If not, why species-lvl identification of pathogens is so important? Please, clarify it in the Introduction section.

Please, involve potential implementation of the presented results in tea cultivation and plant protection.

Response: Thank you for your constructive comments. Actually, we want to express that identifying the specific pathogenic fungi causing tea anthracnose can help the researchers pinpoint the specific pathogen, and finding the accurate pathogen is the prerequisite for studying the anthracnose resistant of tea plant. So this sentence is ambiguous and extend too much research meaning. Among the former researches, tea anthracnose, tea leaf blight and other alike diseases will be confused and the pathogens of anthracnose are different in various areas, which leads to suspect results for molecular studies of tea disease resistance. Our aim is to find the right pathogens causing tea anthracnose. Above all, we have changed this sentence into “Identification of specific pathogenic fungi causing tea anthracnose is a prerequisite for studying the resistance of tea plants to anthracnose” to express our goal more accurate.

In addition, we have added several discussion about present results in tea cultivation and plant protection in “Introduction” and “3.3. Interaction of Dt and Colletotrichum”.

2) "To explore the composition and function watery lesions may be a meaningful research in the future. "

This sentence should be placed in the Discussion section and then extended.

Response: Thank you for your comments. We have added a paragraph in discussion “3.1. Phenotype and Isolated Main Fungi of Diseased Tea Leaves”.

3) Please clarify, how many plant individuals (not leaves) were studied? How much the results differed between different plant individuals within and/or between three study sites?

Response: 

According to your instruction, we have added the sampling details in “4.1 Collection and Isolation”. Actually, we chose three representative tea gardens. Among each tea garden, we searched the diseased tea leaves fitting the symptoms as P1, P2 and P3, then we chose more than three tea plant individuals and sampling more than three diseased tea leaves with equivalent leaf position randomly in each individuals. Moreover, we sampling the diseased leaves in June, July and August continuously with two years. Above all, we at least sampling the leaves more than 6 times, 18 tea plant individuals and 54 leaves.

Our experiment lasted two years. At the beginning of the experiment, we attempted to find differences between three sampling sites. But we didn't find differences on appearance of disease spot or isolates. Thus, we focus on the relationship between acervuli and spores, strain identification in the subsequent experiment.

4) Lines 430-431

"Tea leaves were collected from Camellia sinensis cv. Longjing43 (LJ43) in three different tea gardens of Zhejiang Province, China."

Are only three sites enough to study the participation of each pathogenic fungal species in tea leaves in tea plantations in general?  Please, explain it using well-published studies.

Response: Thank you for your constructive comments. This is a rather important issue in this study. Our group have researched tea anthracnose for several years [1, 2]. We have spotted Dt using stereoscope in previous studies several times and assumed it may be another main pathogen in tea anthracnose, but we didn’t study Dt systematically before. In this study, our principle goal is to identify whether Dt is the main pathogen causing tea anthracnose or not. Besides, we want to verify its interaction with Colletotrichum. That’s why we tested the pathogenicity of Dt and co-incubated it with Colletotrichum. To assure its reliability, we sampled the representative diseased leaves in continuously months and years, and the results still stood the same. As we know, if we focus on the pathogens species distribution and variety, we should sample the diseased tea leaves as many as possible, but we aim to study the characteristics and pathogenicity of Dt. In addition, after we colonize the Dt on tea leaves, the disease symptoms were consistent with field observation. Combined with above information, we consider Dt as a main pathogen causing tea anthracnose is of universal applicability.

Reference:

[1] Lu, Q.; Wang, Y.; Li, N.; Ni, D.; Yang, Y.; Wang, X. Differences in the characteristics and pathogenicity of Colletotrichum camelliae and C. fructicola Isolated from the tea plant (Camellia sinensis (L.) O. Kuntze). Front. Microbiol. 2018, 9, 3060.

[2] Wang Y, Hao X, Lu Q, Wang L, Qian W, Li N, Ding C, Wang X, Yang Y. Transcriptional analysis and histochemistry reveal that hypersensitive cell death and H2O2 have crucial roles in the resistance of tea plant (Camellia sinensis (L.) O. Kuntze) to anthracnose. Hortic Res. 2018, 5,18

5) Please, modify the title, as follows:

Characteristics of pathogenic fungi responsible for anthracnose of Camellia sinensis leaves in three tea gardens located in Zhejiang Province in China.

Response: Thank you for your constructive comments. As the query you raised before, we think the main purpose of this study focused on the characteristics and pathogenicity of Discula theae-sinensis. If possible, we suggest the title remains the former one.

6) Supplementary files

All references should be added to the Reference list.

Response: Thank you for your comments. We have added all references in the reference list.

7) Conclusions

Line 425

"The main cause of anthracnose might be the concerted action of a variety of fungi."

Please, expand this sentence and place it as the main conclusion of the study in the Abstract and Conclusion sections.

Response: Thank you for your comments. Based on your comments, we regard this paragraph of results is redundant. So we rearrange the expression and put them into Abstract and Conclusion.

Reviewer 4 Report

For this article, I recommend a substantial rewriting and reevaluation of the data relevant to the composition. The text is excessively lengthy, featuring an excessive number of figures, and the presentation of data is characterized by confusion. Moreover, the bibliography is nearly absent (above all in the Discussion section), and the Discussion section is notably brief.

I suggest some changes below, but the text needs to be completely changed.

Introduction:

The Introduction section is very confusing and does not convey clear information. It reduces to a mere list of fungal taxa, probably causal agents of anthracnose. It needs a complete restructuring and revision. I provide some example to improve the text:

Line 36-37, please rewrite, the sentence is not clear

Line 38, please change the sentence, for example” the period of maximum diffusion is from Abril..”

Line 39-40, this sentence is not coherent with the previous paragraph, please rewrite.                  

Line 55, change the word “distinct”.

Line 60, you have to specify that Camellia is the tea plant.

Line 63, the sentence is not clear.

Line 67-68, please clarify the taxonomic order, it is very confused among genus, phylum and order.

Line 68, the sentence seems incomplete, please add more information.

Line 71, please change the sentence, for example “Among them, C. camelliae (Cc) and C. fructicola (Cf) have been characterized as the principal causal agents of tea anthracnose, based on geographic distribution and strain quantity.”

Materials and Methods:

Line 432, “in terms of tea disease profile” please specify, it is not clear.

Line 437, Did you check the leaves sterilization? Please add this information.

Line 441-442, Which experiments? Please specify.

Line 472, “non wounded”.

Results:

All the sections are too long. It is impossible for the reader to remember and understand all this information. Please reduce the number of figures/graphs and above all discard irrelevant information/data.

Section 2.2, it is confused and not clear. Why did you decide to describe only some isolates?

Figure 2, please move to supplementary materials

Figure 3, Why did you choose this isolates? It is not explained, please add information.

Figures 4-5-6, Why did you choose this isolates? It is not explained, please add information. Now from 11 isolates of the previous figure we can see the morphological characteristic of only 8 of them, why?

Discussion:

Please report the number of the figures of the Results section, it is impossible to follow the text.

Line 337, it is the first time that you talk about different sampling times. Please add information in Material & Methods and in Results sections.

Line 352, which studies? Please add references.

Does more than 13 pages of results fit into less than two pages of discussions?

Please, add more information/bibliography/hypotheses to the Discussions and drastically cut the Results.

The text needs to be re-editing, the language is confusing in many points.

Author Response

Dear editor/reviewer,

We express our gratitude for your valuable comments. We have revised the manuscript, and would like to re-submit it for your consideration. All comments raised by the reviewers were addressed, and the amendments are highlighted in yellow in the revised manuscript. Also, the revised manuscript has been re-edited by a native English speaker from an editing company. Point by point response to the reviewers’ comments are listed in details below.

Your sincerely,

Lu Qinhua

Replies to Reviewer 4

1) Line 36-37, please rewrite, the sentence is not clear

Response: According to your suggestion, we have rewritten this sentence as “Anthracnose is a severe and wide spread disease which affects the growing status of tea plant and reduce the quality of tea product”.

Line 38, please change the sentence, for example” the period of maximum diffusion is from Abril..”

Response: According to your suggestion, we have rewritten this sentence.

Line 39-40, this sentence is not coherent with the previous paragraph, please rewrite.

Response: Thank you for your comments. We write this sentence to support the influence of tea anthracnose in tea industry. However, it is unsuitable in this paragraph. So we have deleted this sentence to make this paragraph more readable.

Line 55, change the word “distinct”.

Response: Thank you for your comments. We have rewritten this sentence and rearranged this paragraph.

Line 60, you have to specify that Camellia is the tea plant.

Response: Thank you for your comments. We have added more information of this tea plant.

Line 63, the sentence is not clear.

Response: According to your comments, we have changed this sentence and moved part of this sentence to the forth paragraph of the Introduction.

Line 67-68, please clarify the taxonomic order, it is very confused among genus, phylum and order.

Response: Based on your this query and third query, we have deleted the redundant content and rearrange the taxonomic order and the figures (figure 2 and figure 4).

Line 68, the sentence seems incomplete, please add more information.

Response: According to your instruction, we think this sentence is improper in this part. So we deleted it and added such information into discussion.

Line 71, please change the sentence, for example “Among them, C. camelliae (Cc) and C. fructicola (Cf) have been characterized as the principal causal agents of tea anthracnose, based on geographic distribution and strain quantity.”

Response: Thank you for your constructive comments. We have undergone major revisions in introduction. All mentioned problems were modified. 

2) Materials and Methods:

Line 432, “in terms of tea disease profile” please specify, it is not clear.

Response: Thank you for your constructive comments. This problem was modified. Diseased samples were identified and collected from leaves with visible anthracnose symptoms which are subround or irregular scabs, scattered small black sporophores, water-soaked lesions and the absence of thin ring spots

Line 437, Did you check the leaves sterilization? Please add this information.

Response: Thank you for your comments, to check the efficacy of this method of surface sterilization, 100 µL of the last wash water was incubated on PDA plates. This sentence was added in 4.1Collection and Isolation.

Line 441-442, Which experiments? Please specify.

Response: Thank you for your constructive comments. This problem was modified. In this section, we did the experiments, including measurement of growth rate, observation colony and conidial characteristics and induction appressoria.

Line 472, “non wounded”.

Response: Based on your comments, all mentioned problems were modified.

3) All the sections are too long. It is impossible for the reader to remember and understand all this information. Please reduce the number of figures/graphs and above all discard irrelevant information/data.

Response: Thank you for your constructive comments. Our whole study is focused on the Discula theae-sinensis, to scrupulously operate this study, we decided to delete the other species in Diaporthales besides Discula theae-sinensis in the manuscript. In addition, we deleted the figure 3 and figure 10, merged figure 4, 5, 6, changed the form of figure 7, simplified figure 8, redo the figure 9. Besides, we added some tables for readers to understand easily.

4) Section 2.2, it is confused and not clear. Why did you decide to describe only some isolates?

Response: The primary aim of the study was to clarify the taxonomy and phylogeny of Diaporthales species. And then, morphological characters of Diaporthales isolates were displayed. After we made the taxonomy of Diaporthales, the strains could be divided into three major clusters, the chosen isolates were representative in each cluster. In the revised version, we only focused on Discula theae-sinensis, and redo the taxonomy. Now, we chose DX7 and DX14 to study their morphological characters, because they were representative in each cluster.

5) Figure 2, please move to supplementary materials

Response: Discula theae-sinensis identification is an important part of present study. Figure 2 was used to show results of molecular identification. For a long time, the taxonomy of Discula theae-sinensis species was unclear. In recent years, DNA data were used to identify Discula theae-sinensis and improve accuracy of classification. Thus, we think Figure 2 is important for this study and we suggest it remains in main text.

6) Figure 3, Why did you choose this isolates? It is not explained, please add information.Figures 4-5-6, Why did you choose this isolates? It is not explained, please add information. Now from 11 isolates of the previous figure we can see the morphological characteristic of only 8 of them, why?

Response: Thank you for your constructive comments. These statements mention a common problem that how to choose isolates from each species. We chose the representative isolates randomly within the species which from different clusters. It is our mistake that some data were not shown in the manuscript. In revised manuscript, the number of Discula theae-sinensis in the figures and morphological characteristic is consistent in quantity.

7) Discussion:

Please report the number of the figures of the Results section, it is impossible to follow the text.

Line 337, it is the first time that you talk about different sampling times. Please add information in Material & Methods and in Results sections.

Response: Thank you for your constructive comments. We have added the sampling times in the 2.1. Phenotypic Characterization of Collected Tea Leaves and 4.1. Collection and Isolation to make the manuscript more readable.

Line 352, which studies? Please add references.

Response: According to your comments, we have rewritten the sentence and added the references in revised version. 

Does more than 13 pages of results fit into less than two pages of discussions?

Please, add more information/bibliography/hypotheses to the Discussions and drastically cut the Results.

Response: Thank you for your constructive comments. We have rewritten most part of the Discussions and deleted redundant content in the results. Because we put many figures in the results, it seems to be long, and some figure were deleted and merged now. In the revised version, the results have about 1700 words and the discussions contains about 1300 words, the length are relatively balanced.

Reviewer 5 Report

Dear Authors,

The MS “Characteristics and Pathogenicity of Discula theae-sinensis Causing Camellia sinensis Anthracnose, and Synergistic Effect with Colletotrichum spp.” by Li et al. needs to improve many sections and this ms should be clear from the beginning to understand what the new contribution to the topic is. In addition, the M&M should be more clear.

Thanks

Author Response

Dear editor/reviewer,

We express our gratitude for your valuable comments. We have revised the manuscript, and would like to re-submit it for your consideration. All comments raised by the reviewers were addressed, and the amendments are highlighted in yellow in the revised manuscript. Also, the revised manuscript has been re-edited by a native English speaker from an editing company. Point by point response to the reviewers’ comments are listed in details below.

Your sincerely,

Lu Qinhua

Replies to Reviewer 5:

Thanks for the carefully review, some spelling, case mistakes have been rectified.

1) Line 28 The keywords arrange in alphabetical order

Response: Thank you for your constructive comments. It has been modified as requested.

2) Line 31-37; Line 41-44; Line 45 please find reference to support.

Response: Thank you for your constructive comments. It has been modified as requested.

3) Line 68 Melanconiales is wrong

Response: We have rechecked it. And according to reference [1], we think Melanconiales is correct. In the introduction of this reference, the authors wrote:

“The form-genus Colletotrichum Corda (form-order Melanconiales; form-class Coelomycetes; subdivision Deuteromycotina) comprises imperfect fungal species which exist as Glomerella (subdivision Ascomycotina) in their sexual, teleomorphic or perfect state.”

Reference: 

[1] Latunde-Dada, A.O. Colletotrichum: tales of forcible entry, stealth, transient confinement and breakout. Mol Plant Pathol. 2001, 12, 187-198

4) Line 72 please change reference

Response: We have rechecked and modified.

5) Some result can be brief, as some are too long without important information.

Response: Thank you for your comment. Combined with the third query of the Reviewer 4, and our whole study is focused on the Discula theae-sinensis, to scrupulously operate this study, we decided to delete the other species in Diaporthales besides Discula theae-sinensis in the manuscript. In addition, we have rewritten the results and discussions to make the manuscript more readable.

6) Line 128 Lopadostoma polynesium should be ltalic. 0.05 is wrong

Response: Thank you for your comment. It is our mistake that mistyped the number. Now the figure and figure legends were amended and the data was also checked.

7) The method of growth rate didn’t mention in M&M

Response: We have rechecked the whole manuscript and added measuring method of growth rate in 4.2. Morphological Characterization.

8) Better to use same type of phylogenic tree

Response: According to your comments, we have redo the phylogenic tree of former figure 7 and make the phylogenic trees in the manuscript consistent.

9) line 214 C. tropicicola should use a full name

Response: We have rechecked the whole manuscript and corrected them.

10) Line 369-375 more likely introduction

Response: According to your comments, we have deleted this part in the Discussions and rewritten most part of the Discussions.

11) Line 429 please mention the original code that you use for those isolates

Response: Thank you for your comment. Before we prepared the manuscript, we used the original code of those isolates, while it became hard to remember for the readers. To simplified the names, we chose to use new simple names with abbreviation and symbol. In addition, we uploaded sequences to NCBI in order to get GenBank accession numbers and listed them in table S2 and S3. The tables would help readers to find the original codes of the isolates.

12) Line 445 please metion how old of mycelium, culture in which media

Response: Thank you for your comment. We have updated information about condition of culture and growth time of mycelium in 4.2. Morphological Characterization.

13) Line 471 All isolates tested and how about control?

Response: In this study, three replicates were included for each treatment. Sterile distilled PDA discs with no mycelia were used as controls. We have added above-mentioned information in 4.5 Pathogenicity Tests of Discula theae-sinensis.

14) Line 490-500 please rewrite as it’s difficult for the reader or make the diagram

Response: Thank you for your constructive comments. It has been modified as requested. We have added table 1 to make it easier to understand.

Round 2

Reviewer 4 Report

The changes made to the text improved its quality and facilitated the reader's understanding. I think the text is ready to be published.

Author Response

Dear reviewer,

Thank you for your comments!

According to your suggestions, we have amended the results and improved our work to be acceptable.

Thank you again!

Sincerely,

Li Qingsheng

Reviewer 5 Report

Dear Editor(s) and authors,

I strongly suggest the author check the classification of Colletotrichum with the recent publication (especially check from taxonomic paper).

Thanks

Author Response

Dear reviewer,

Thank you for your comments, and kindly reminder.

We have recheck the classification of Colletotrichum and replaced the reference in Line 54-55.

I am really sorry for the mistake we made.

Thank you again!

Sincerely,

Li Qingsheng